# The Serotonin Syndrome: From Molecular Mechanisms to Clinical Practice

**DOI:** 10.3390/ijms20092288

**Published:** 2019-05-09

**Authors:** James Francescangeli, Kunal Karamchandani, Meghan Powell, Anthony Bonavia

**Affiliations:** Department of Anesthesiology and Perioperative Medicine, Penn State Milton S. Hershey Medical Center, Hershey, PA 17033, USA; jfrancescangeli@pennstatehealth.psu.edu (J.F.); kkaramchandani@pennstatehealth.psu.edu (K.K.); mpowell2@pennstatehealth.psu.edu (M.P.)

**Keywords:** serotonin syndrome, polypharmacy, 5-hydroxytryptophan, antidepressants, genetic polymorphisms

## Abstract

The serotonin syndrome is a medication-induced condition resulting from serotonergic hyperactivity, usually involving antidepressant medications. As the number of patients experiencing medically-treated major depressive disorder increases, so does the population at risk for experiencing serotonin syndrome. Excessive synaptic stimulation of 5-HT_2A_ receptors results in autonomic and neuromuscular aberrations with potentially life-threatening consequences. In this review, we will outline the molecular basis of the disease and describe how pharmacologic agents that are in common clinical use can interfere with normal serotonergic pathways to result in a potentially fatal outcome. Given that serotonin syndrome can imitate other clinical conditions, an understanding of the molecular context of this condition is essential for its detection and in order to prevent rapid clinical deterioration.

## 1. Introduction

The serotonin syndrome (SS) is a clinical condition resulting from serotonergic over-activity at synapses of the central and peripheral nervous systems. The true incidence of the disease is unknown, given that its severity varies and that many of its symptoms may be common to other clinical conditions. The SS is triggered by therapeutic drugs that are not only common, but ones whose use appears to be increasing at an alarming rate [1]. The most common drug triggers of SS are antidepressants, for which the incidence of use in adults in the United States has increased from 6% in 1999 to 10.4% in 2010 [2]. Furthermore, reported ingestions of selective serotonin reuptake inhibitors (SSRIs) increased by almost 15% from 2002 to 2005 [3,4,5,6,7].

In this review article, we will describe the clinical pathophysiology of SS and present the leading molecular theories underlying the disease. Being a purely clinical diagnosis with protean manifestations, an understanding of the presentation of this disease and related confounding diagnoses is necessary to establish a relevant context. What follows is the putative molecular basis of the disease, and how certain genetic polymorphisms are hypothesized to contribute to its manifestations in predisposed individuals. This information will facilitate an understanding of how certain medications can trigger the syndrome, especially in high risk patients. It will also help readers to comprehend current treatment strategies and directions for future research in the field.

## 2. Clinical Context

### 2.1. Definition and Epidemiology

The diagnostic basis of SS includes the triad of altered mental status, autonomic hyperactivity, and neuromuscular abnormalities [8,9] in patients exposed to any medication which increases the activation of serotonin (5-hydroxytryptamine; 5-HT) receptors in the body [10]. These medications include SSRIs, monoamine oxidase inhibitors (MAOI), opioid analgesics, antiemetics, illicit drugs, and others [10]. The widespread use of these medications puts a large portion of the population at risk for developing this disease, especially if used in combination [11]. A retrospective cohort study reviewing Veterans Health Administration records from 2009–2013 showed a disease incidence of 0.23% in patients exposed to serotonergic medications (0.07% overall) [12]. This same study also reported a 0.09% incidence of SS in commercially insured patients exposed to serotonergic medications in 2013 (0.03% overall) [12]. The median cost per inpatient hospital stay associated with SS was $10,792 for commercially insured patients and $8765 for Veterans Health Administration patients [12]. No associated mortality data was reported [10,12]. There are no reports identifying specific patient risk factors that are associated with the development of SS outside of the genetic polymorphisms that will be described in further detail later. However, many of the medications with the potential to cause SS are commonly used in the geriatric population, thus placing these patients at higher risk of developing the disorder.

Since SS varies in presentation, it is likely to be grossly underdiagnosed in clinical practice and, thus, studies into its precise mechanism are very limited [13]. Much of the relevant research data is derived from animal models and from case descriptions of individuals in whom the disease has been highly suspected.

### 2.2. Manifestations and Diagnosis

Several diagnostic algorithms have been proposed since SS was first recognized as a discrete disease entity (Table 1). The main challenges encountered in establishing formal diagnostic criteria are (1) the wide range of symptoms exhibited by patients affected by the disease and (2) the lack of a confirmatory laboratory test. Thus, the diagnosis of SS remains purely clinical at present. The first diagnostic criteria were proposed by Sternbach et al. in 1991, based on a review of 38 published case reports in which patients demonstrated several shared symptoms [8]. Cases were reported by as many as 12 different investigators, and the most commonly reported symptoms included confusion (*n* = 16), hypomania (*n* = 8), restlessness (*n* = 17), and myoclonus (*n* = 13) [8]. Sternbach’s criteria were based on the inclusion of three or more of the most commonly noted symptoms extracted from the 38 cases. The major weakness of Sternbach’s criteria was the inclusion of four separate altered mentation symptoms (confusion/hypomania, agitation, and incoordination), which made it possible to diagnose SS purely based on mental status changes [11]. Such mental status changes could be commonly observed in many other conditions such as alcohol and drug withdrawal states and anticholinergic delirium [3], a limitation which Sternbach fully acknowledged.

Between 1995 and 2000, Radomski and colleagues [14] reviewed subsequent cases of suspected SS with the goals of refining Sternbach’s diagnostic criteria and outlining the medical management of this disorder. The most recent diagnostic criteria, however, were developed by Dunkley et al. in 2003 [11]. Dunkley’s criteria were formed through the use of a toxicology database called the Hunter Area Toxicology Service, which included patients who were known to have overdosed on at least one serotonergic medication. A decision tree was constructed by including symptoms which recurred at a statistically significant frequency in patients with SS that had been diagnosed by a medical toxicologist. This diagnostic algorithm was both more sensitive (84% vs. 75%) and more specific (97% vs. 96%) in diagnosing SS than Sternbach’s criteria [11]. The Hunter Serotonin Toxicity Criteria, as they are now known, are generally considered the gold standard for diagnosing this disease [10]. They consist of the aforementioned triad of altered mental status, neuromuscular excitation and autonomic dysfunction. Symptoms usually occur within one hour of exposure to triggering medications in 30% of patients, and within six hours in 60% of patients [1]. Mild cases may present as little more than flu-like symptoms, while severe cases may progress rapidly to cardiovascular collapse and death (Figure 1).

### 2.3. Differential Diagnosis

Several potentially life-threatening diseases share signs and symptoms similar to those present in SS, making the importance of an accurate and timely diagnosis imperative (Table 2). These diseases include neuroleptic malignant syndrome, anticholinergic toxicity, malignant hyperthermia [10], antidepressant discontinuation syndrome, and alcohol withdrawal. All may result in some degree of autonomic dysregulation (including tachycardia, hypertension, and hyperthermia) and an acutely altered mental status [10]. The first three of these disorders are most closely related, and their defining clinical features are summarized in Table 2 [10,15]. Neuroleptic malignant syndrome is typically associated with the use of antipsychotic medications, such as dopamine antagonists, and presents with signs of muscular rigidity. These signs typically occur several days following exposure to triggering medications, unlike SS which immediately follows medication exposure [15]. Anticholinergic toxicity, as the name implies, is associated with the use of anticholinergic medications. Typical signs of this disorder include dry, hot skin and absent bowel sounds, contrasting with the diaphoresis and hyperactive bowel sounds that are typical of SS [10]. Malignant hyperthermia is associated with exposure to volatile anesthetic agents or to the depolarizing neuromuscular blocker, succinylcholine. The result is severe muscle rigidity and hyporeflexia [15].

Like alcohol withdrawal, benzodiazepine or barbiturate withdrawal can cause a hyperactive state which may be mistaken for SS. Alcohol, benzodiazepines, and barbiturates are central nervous system depressants, and their abrupt discontinuation in dependent patients may trigger a potentially life-threatening reaction. Alcohol withdrawal typically ranges in severity from disorientation and tremulousness to ‘delirium tremens’, which occurs within 48–72 h of the last alcoholic exposure and may be deadly. Delirium tremens, as the name implies, is characterized by disorientation and global confusion, hallucinations, and sometimes seizures [16]. Autonomic disturbances such as tachycardia and hypertension are common in severe cases although, unlike SS, hyperthermia is not [16]. The timing of benzodiazepine or barbiturate withdrawal varies depending on the medication half-life, and longer-acting medications are less likely to cause withdrawal symptoms [16]. Anxiety is common following benzodiazepine or barbiturate withdrawal, although hyperthermia and neuromuscular abnormalities (e.g., clonus and hyperreflexia) are not.

Withdrawal from antidepressant medications may also be mistaken for SS. Antidepressant withdrawal syndrome is typically associated with second-generation antidepressants [17], although it can also occur with SSRIs and tricyclic antidepressants (TCAs) [18]. It usually begins within three days of stopping the offending medication, and symptoms are usually mild and resolve spontaneously within one to two weeks [19]. Antidepressant withdrawal syndrome can cause a spectrum of signs and symptoms including neuromuscular abnormalities such as akathisia, myoclonic jerks, and tremor, as well as altered mental status, psychosis, and mood disturbances. However, it is not typically associated with hyperthermia or with autonomic disturbances such as tachycardia [18].

## 3. The Molecular Basis for the Serotonin Syndrome

### 3.1. Animal Models

The exact pathophysiologic mechanism for SS has been difficult to elucidate, partially due to the multitude of known serotonin receptors classes and subtypes [21]. Although serotonin is the primary ligand for all of these receptors, only the stimulation of certain receptor subtypes leads to SS [21,22,23]. There is a dearth of research on the pathology of serotonin toxicity in humans, necessitating data collection from experimental animal models. Haberzettl et al. analyzed 109 publications in which rodent behavioral and autonomic responses were altered by the administration of serotonin agonists or serotonin-enhancing drugs [24]. Further, they identified “traditional behavioral responses,” a distinct set of rodent responses that are consistently observed following the administration of these medications and, thus, believed to be the rodent equivalent of SS in humans. These include, amongst others, forepaw treading, hindlimb abduction, head weaving, head twitching, back muscle contraction, and hyperthermia [8,22,24,25]. Haberzettl used these behaviors to assess the utility of standardized animal models of SS for the study of this disease in humans.

While the relevance of these rodent models remains questionable, animal models have, nonetheless, provided valuable insights into the importance of 5-HT receptors and the 5-HT transporter in the pathophysiology of SS. Until recently, SS was believed to be primarily a disease involving 5HT_1A_ receptors [26], since the most prominent effects in the rodent serotonin behavioral syndrome appeared to be mediated by postsynaptic 5-HT_1A_ receptors [27,28,29,30,31]. Subsequent animal studies involving hyperthermia and muscle excitation have shed some more light on the potential pathophysiology of SS in humans. In these experiments, rats exposed to various serotonin receptor subtype-selective antagonists demonstrated hyperthermia that was significantly associated with 5-HT_2A_ receptor stimulation [32]. Additionally, hyperthermia was prevented when rats exposed to a serotonin precursor and MAOI was introduced to the 5-HT_2A_ receptor antagonist ritanserin [33]. Conversely, when the 5-HT_1A_ receptor was directly stimulated, rats develop hypothermia [28,34]. Although 5-HT_2A_ stimulation would appear to be the cause of hyperthermia in SS, this is likely an over-simplification. Rat models of SS also show increased levels of norepinephrine, dopamine and glutamate within the hypothalamus [35,36], as well as elevated dopamine and norepinephrine levels in the frontal cortex of rats exposed to 5-HT_2A_ agonists [37]. The N-methyl-D-aspartate (NMDA) antagonist memantine, as well as positive allosteric modulators at GABA_A_, diazepam, and chlormethiazole, have all been found to be effective at decreasing the hyperthermic response in SS model rats. This suggests that, at least in rats, GABA and NMDA receptors are also involved in the pathogenesis of SS [33,38].

### 3.2. Molecular Pathways

An understanding of the putative mechanisms of SS necessitates a basic understanding of serotonin synthesis and clearance. 5-hydroxytryptamine is produced in enterochromaffin cells of the gastrointestinal tract as well as in the midline raphe nuclei of the brainstem. The serotonin produced in enterochromaffin cells is responsible for most of the neurohormone present in the blood, and of the approximately 10 mg of serotonin present in the human body, 4–8 mg is found in enterochromaffin cells located in the gastric and intestinal mucosa [26]. The remainder is found in the central nervous system and in platelets (where it is taken up and stored alone, since platelets do not synthesize serotonin) [26]. The serotonin produced in the gastrointestinal tract stimulates physiologic functions as diverse as vasoconstriction, uterine contraction, bronchoconstriction, gastrointestinal motility, and platelet aggregation. In contrast, centrally-released serotonin inhibits excitatory neurotransmission and modulates wakefulness, attention, affective behavior (anxiety and depression), sexual behavior, appetite, thermoregulation, motor tone, migraine, emesis, nociception, and aggression [10,39]. The signs and symptoms associated with SS include a conglomeration of effects produced by overzealous activation of central and peripheral serotonin receptors.

#### 3.2.1. Synthesis and Release

5-hydroxytryptamine is synthesized from the essential amino acid L-tryptophan, supplied by dietary protein (Figure 2). In the brain, other neutral amino acids, such as phenylalanine, leucine, and methionine, are transported by the same carrier as L-tryptophan [40]. Hence, the entry of tryptophan into brain is related not only to its concentration in blood, but also to its concentration relative to that of these other amino acids [40]. While increased dietary intake of L-tryptophan can lead to SS [41], the therapeutic relevance of amino acids that compete with tryptophan for treatment of symptoms of SS has not yet been explored.

The first, and rate-limiting, step in serotonin synthesis involves the conversion of L-tryptophan to 5-hydroxytryptophan (5-HTP) by the enzyme tryptophan hydroxylase. Inhibition of this reaction results in a marked depletion of serotonin. 5-HTP is then converted to serotonin by the enzyme aromatic L-amino acid decarboxylase [10,42,43]. Once serotonin is synthesized, it is packaged into vesicles via a vesicular monoamine transporter (VMAT). Storage of serotonin in vesicles requires its active molecular transport from the cytoplasm by using energy stored in the electrochemical gradient generated by a vesicular H^+^-ATPase. Thus, uptake of serotonin is coupled to efflux of hydrogen cations. In the central nervous system, these vesicles then release serotonin into the synaptic cleft, most likely through calcium-dependent exocytosis [42,44]. The rate of serotonin release is dependent on the firing rate of serotonergic neurons.

#### 3.2.2. Termination of Effects: Reuptake and Metabolism

The synaptic effects of serotonin are terminated by molecular reuptake and metabolism [42]. Reuptake occurs via the serotonin reuptake transporter (SERT) that is located along the axon of presynaptic serotonergic neurons. Reuptake by the SERT controls the amount of serotonin in the synaptic cleft. It is an energy- and temperature-dependent process and it can be inhibited by drugs such as SSRIs, serotonin noradrenaline reuptake inhibitors (SNRIs), and inhibitors of sodium–potassium ATPase activity [44]. Inhibition of serotonin reuptake leads to accumulation of this molecule within the synaptic cleft, thereby potentially overstimulating its molecular receptors.

There is significant structural homology amongst molecular transporters of serotonin, norepinephrine and dopamine. Furthermore, different drugs exhibit selective but incomplete inhibition of each transporter protein. For example, secondary amine TCAs, such as desipramine, are 25- to 150-fold more potent at inhibiting transport of norepinephrine than serotonin, while SSRIs are 15 to 75 times more potent at inhibiting the uptake of serotonin than the uptake of norepinephrine [40]. Drugs such as cocaine, on the other hand, are nonselective inhibitors of all three transporters. Inhibition of serotonin reuptake is one of the most common mechanisms by which drugs cause SS. 3,4-Methylenedioxy methamphetamine (MDMA), commonly known as the recreational drug ‘ecstasy’, and fenfluramine are unique in that they are able to target both VMATs and SERTs [44]. These drugs directly compete for the VMATs’ substrate binding site, and they also act as SERT substrates inhibiting serotonin transport into the cell and facilitating efflux by the SERT [44]. SERTs are the targets of SSRIs, TCAs, and clomipramine, and long-term exposure to SERT-blocking drugs, such as SSRIs, eventually causes downregulation of the protein transporters [44].

The primary pathway for serotonin metabolism is oxidative deamination by the enzyme monoamine oxidase (MAO). MAO converts serotonin to 5-hydroxyindoleacetaldehyde, and this product is further oxidized by a nicotinamide adenine dinucleotide (NAD)-dependent aldehyde dehydrogenase to form 5-hydroxyindoleacetic acid (5-HIAA). The intermediate acetaldehyde can also be reduced by NADH-dependent aldehyde reductase to form the alcohol 5-hydroxytryptophol [40]. Either oxidation or reduction may take place depending on the concentration ratio of oxidized (NAD^+^) to reduced (NADH) nicotinamide adenine dinucleotide in the tissue. In the brain, 5-HIAA is the primary metabolite of serotonin.

There are at least two isoenzymes of MAO: MAO-A and MAO-B. The metabolism of serotonin and norepinephrine preferentially occurs by MAO-A [44], since MAO-A enzymes are specific for monoamine neurotransmitters and inactivate these molecules by terminal deamination [45]. The first prescribed MAOI were nonselective and irreversible, but more modern drugs in this class are selective for MAO-A or MAO-B and are reversible [46]. MAO inhibitors (MAOIs), therefore, can increase the concentration of serotonin and contribute to the development of SS. MAOIs that are irreversible, nonselective, or inhibit MAO-A are strongly associated with severe cases of SS, especially when these agents are used in combination with an SSRI or a serotonin-releasing drug (e.g., meperidine, dextromethorphan, and MDMA) [47,48,49,50,51].

#### 3.2.3. Serotonin Receptors Subtypes

Serotonin receptors consist of seven numerically named families, 5-HT_1_ through 5-HT_7_, several of which have multiple members (e.g., 5-HT_1A_, 5-HT_1B_, 5-HT_1C_, 5-HT_1D_, 5-HT_1E_, and 5-HT_1F_) [10,42]. This classification scheme is based on a combination of the pharmacologic properties of each receptor, their secondary messenger systems, and their amino acid sequence [43]. 5-HT_1_, 5-HT_2_, 5-HT_4,_ 5-HT_6_, and 5-HT_7_ are all G-protein coupled receptors, while 5-HT_3_ is a ligand-gated receptor [44]. Regarding secondary messenger systems, the 5-HT_1_ family inhibits adenylate cyclase while the 5-HT_2_ family activates phospholipase C. [43] Each of the receptor subtypes is present in the central nervous system, while subtypes 1C, 2, 3, and 4 have additionally been identified in the gastrointestinal tract [43].

Although no single receptor has been identified as causing the signs and symptoms associated with SS, it appears that 5-HT_2_ receptor agonism and, more specifically, 5-HT_2A_ agonism plays a significant role in causing the severe symptoms associated with serotonin toxicity [20,52]. 5-HT_2A_ is present in the cerebral cortex, gastrointestinal tract, vascular and bronchial smooth muscle, and platelets [43], and causes neuroexcitation, bronchoconstriction, platelet aggregation, and smooth muscle contraction. The role of genetic polymorphisms affecting the 5-HT_2A_ receptor is the focus of current research in neuropsychiatric disorders [20,53]. It may also shed light on molecular basis of SS, and will be discussed in more detail later.

### 3.3. Medications Triggering the Serotonin Syndrome

Several discrete mechanisms have been hypothesized to explain how drugs cause SS, although it is likely that more than one of these mechanisms may need to be simultaneously triggered in order to result in clinically significant SS. Thus, the most common cause of SS is polypharmacy causing (1) inhibition of serotonin uptake, (2) decreased serotonin metabolism, (3) increased serotonin synthesis, (4) increased serotonin release, and/or (5) activation of serotonergic receptors [22]. Given the centrality of serotonin excess in precipitating the disease, the term ‘serotonin toxicity’ has been advocated by some in lieu of ‘serotonin syndrome,’ in order to more accurately describe the disease pathophysiology [52]. Table 3 provides a summary of the drugs that have been reported to cause serotonin syndrome. The combination of an MAOI with an SSRI, and SNRI or another MAOI is the most dangerous combination and the most likely to result in SS [54]. The Therapeutic Goods Administration of Australia published a drug safety report in 2009, warning against the risk of life-threatening reactions when MAOIs are combined with SSRIs or phenylpiperidine opioids [55].

Increased production of serotonin (through an overabundance of the precursor molecule L-tryptophan [58]) as well as decreased metabolism of serotonin (by the administration of MAOI [59]), may each cause receptor overstimulation and SS. The combination of L-tryptophan administration with the use of the MAO-A inhibitor clorgyline has been reported to be sufficient to trigger SS in experimental rats [23,33,35,36,38]. Serotonin metabolism may also be influenced by alterations in the function of the cytochromes p450 (CYPs). These are enzymatic hemeproteins which generally act as terminal oxidases in electron transfer chains. CYPs are ubiquitous in nature, although in humans they are associated with the inner membranes of endoplasmic reticulum or mitochondria and they account for approximately 75% of drug metabolism [60]. SSRIs (most notably fluoxetine and paroxetine) and antibiotics, such as ciprofloxacin [61] and fluconazole [62], strongly inhibit CYPs such as CYP2D6 and CYP3A4 [63]. These CYPs are responsible for the metabolism of SSRI and other serotonergic drugs such as venlafaxine, citalopram, methadone, tramadol, oxycodone, risperidone, dextromethorphan, and phentermine, resulting in a potentially toxic level of serotonergic drugs as well as a vicious cycle of progressive SSRI accumulation.

Some medications that cause SS act by increasing serotonin concentrations at the synapse without altering the synthesis or metabolism of neurotransmitters. Stimulants such as amphetamines, phentermine, MDMA, and phenanthrene opioids (e.g., morphine, codeine, hydromorphone, buprenorphine, and oxycodone) have been associated with increased localized release of serotonin at presynaptic nerve terminals [59,64,65,66].

Opioid analgesics include some of the most commonly prescribed drugs in the hospital and outpatient settings [67]. They can cause SS when taken in conjunction with other serotonergic medications (such as MAOIs, SSRIs, SNRIs, and TCAs [58]), when taken with serotonin reuptake blockers (such as methadone and tramadol [59]) or when administered alone in high doses. Meperidine, tramadol, methadone and other synthetic opioids have been shown to inhibit SERT function in vitro, although fentanyl and phenantrene opioids such as morphine have not [56]. This implies that opioids have SERT-independent effects in vivo. Seminal animal studies by Tao et al. have provided important insight regarding the opioid-induced modulation of serotonin efflux in the central nervous system [68]. They demonstrated that different opioid subtypes cause distinct serotonergic effects. Whereas the µ- and δ-opioid agonists cause a release of serotonin by the dorsal raphe nucleus, they had no effect on serotonin release in the median raphe nucleus or in the nucleus accumbens [68]. In contrast, κ-opioid agonists produced localized decreases in serotonin in all three regions of the brain. The investigators further discovered that morphine (a predominantly µ-opioid agonist) does not directly stimulate serotonergic neurons [69]. However, it indirectly stimulates serotonin release via activation of opioid receptors on gamma-amino butyric acid (GABA)-ergic and glutamatergic afferent neurons in the dorsal raphe nucleus [70].

The increasing use of opioids along with the increase in serotonergic drug prescriptions in the general population [71] has dramatically increased the risk of SS. In 2016, the food and drug administration (FDA) issued a drug safety communication concerning the risk of serotonin toxicity with the use of opioids [72]. Although there is limited clinical data to support this warning, several cases of opioid-associated serotonin toxicity have been reported [56,73]. In a retrospective analysis of 203 cases of serotonin toxicity registered in the French pharmacovigilance database from 1985 to 2013 [74], for example, the association between opioids and either SSRI or MAOI was implicated in over 25% of the cases of SS. Clinical case reports of opioid-induced serotonin toxicity have often come under heavy criticism relating to inconsistent reporting of toxic events, failure to report important positive and negative findings, and the presentation of incomplete or erroneous information on drugs involved, symptoms, and treatments [75]. Nonetheless, all opioids should be used with caution in patients taking serotonergic drugs.

Lastly, certain medications have serotonin receptor agonist properties and are able to activate serotonin receptors in the absence of endogenous agonist [59]. Buspirone, an anxiolytic that is in common clinical use, has receptor agonist activity at several serotonin receptors including 5-HT_1A_ and 5-HT_2A_ [76]. The drug of abuse, lysergic acid diethylamide (LSD), has also been proposed to act in similar fashion as it independently binds to 5-HT_1_ and 5-HT_2_ receptors [59,77]. Lithium is used for the treatment of certain psychiatric disorders such as bipolar disorder and increases the sensitivity of the postsynaptic serotonin receptors to serotonin without affecting the concentration of serotonin itself [21]. While its precise mechanism of action remains unknown [21,78], its use has frequently been associated with the development of SS [79,80].

### 3.4. Genetic Polymorphisms

It appears that certain individuals with known polymorphisms at the T102C site of the 5-HT_2A_ receptor gene may be predisposed to developing SS [20,53]. Over the last decade, genetic polymorphisms affecting the 5-HT_2A_ receptor have also been implicated in antidepressant therapy failure and in the pathophysiology of neuropsychiatric disorders, ranging from schizophrenia to affective disorders [81]. A prospective, double-blind, randomized pharmacogenetic study compared treatment outcomes with the SSRI paroxetine and the non-SSRI antidepressant mirtazapine in patients having different T/C single nucleotide polymorphisms affecting the 5-HT_2A_ receptor [82]. The study found that patients who are homozygous for polymorphisms at the *HTR2A* locus (C/C) are more likely to discontinue paroxetine due to more severe adverse side effects. Incidentally, mirtazapine has a unique mechanism of enhancing serotonergic and noradrenergic pathways in the central nervous system [83]. It inhibits presynaptic inhibitory receptors on noradrenergic and serotonergic neurons (thus, increasing release of these neurotransmitters in the synaptic cleft). However, since it also blocks 5-HT_2_ and 5-HT_3_ receptors, only serotonergic transmission via 5-HT_1A_ is enhanced [84].

Another published case report described a patient taking the MAOI phenelzine, who developed SS without being exposed to other serotonergic agents. He was subsequently found to be a homozygous carrier for T102C allele (i.e., C/C) [85]. Contrary evidence is presented by Cooper et al., who failed to find a significant increase in clinically significant cases of SS in individuals having polymorphisms at the T102C locus [53].

Individual variations in serotonin metabolism by CYPs have also been proposed to contribute to SS susceptibility [86,87,88]. One case report describes the development of SS in an individual who was taking the SSRI paroxetine in the absence of other known serotonergic medications [87]. While paroxetine infrequently causes SS in isolation, this patient was found to have a polymorphism for the CYP2D6 allele, which may have impaired the metabolism of paroxetine and contributed to the development of SS [87]. A similar case report postulated that altered drug pharmacokinetics may have contributed to SS in a patient taking fluoxetine who was found to have a nonfunctioning CYP2D6 genotype, as well as being heterozygous for an allele of CYP2C19 that results in poor metabolizing ability [88].

The contribution of CYP polymorphisms to the development of SS is further complicated when one considers the multitude of pharmacologic CYP inducers and inhibitors in clinical use today. Medications for the treatment of human immunodeficiency virus (HIV) are notorious for altering the intrinsic metabolic rates of CYPs. Indeed, one case describes the development of SS in an HIV-infected patient taking antiretroviral medications which are known inhibitors of CYP2C19 and CYP3A4. In addition to polypharmacy-induced alterations in CYP-driven drug metabolism, this patient was also found to express a poor metabolizer phenotype of CYP2D6 [86]. Unlike T102C polymorphisms, the CYP2D6 genotype has not yet been shown to alter tolerance to antidepressant medications, although large-scale studies are needed to establish the risk conferred by CYP polymorphisms on the development of SS [83].

As described above, SERT proteins are critical to the termination of synaptic serotonergic activity. Animal studies have suggested that genetic differences in the SERT gene may also partially explain the susceptibility of certain individuals to develop SS [83,89]. Fox et al. showed that SERT knockout mice (SERT^-/-^) exhibited increased susceptibility to SS-like behavior when given serotonergic drugs. Some of these mice even displayed SS-like behavior without administration of 5-HTP. This was also true in mice heterozygous for the gene (SERT^+/-^). However, in wild type mice (SERT^+/+^), administration of both 5-HTP and MAO-A-selective inhibitor was needed to induce SS-like behavior. Furthermore, knockout and heterozygous mice expressed significantly less presynaptic inhibitory Htr1a autoreceptors. These receptors direct a negative feedback mechanism, so that their activation by serotonin decreases serotonin synthesis and, conversely, fewer inhibitory receptors lead to increased serotonin synthesis.

These findings in mice may have implications for patient care. SERT polymorphisms are known to exist in humans [90], and some can reduce SERT function by as much as 50% of normal levels. Further research is needed to assess whether they have a clinically meaningful impact in patients who develop SS.

## 4. Receptor-Targeted Therapy for Serotonin Syndrome

Ideally, the occurrence of SS is prevented by clinicians who are vigilant of patients taking high-risk medications. Preventative strategies include yearly reassessment of the continued need for serotonergic medications, the use of the lowest medication dose required to treat symptoms, accurate medical history-taking with an emphasis on illicit drug use, close medical follow-up during medication dose increases and during hospital encounters where patients may be exposed to triggering medications, and patient education geared at recognizing early symptoms of SS [54].

In patients experiencing SS, toxicity usually resolves following the discontinuation of serotonergic medications. Seventy percent of patients recover completely within 24 h, 40% of patients require admission to an intensive care unit, and only 25% of patients require endotracheal intubation [91]. In addition to stopping serotonergic medications, the treatment of SS in humans has consisted primarily of supportive care, including the administration of benzodiazepines [1,10,92,93,94]. Given that life-threatening manifestations of SS such as rigidity and hyperthermia seem to result only from stimulation of 5-HT_2A_ receptors [20], it would follow that directed therapy for this disease would target this particular receptor. While the 5-HT_2A_ blockers ritanserin and pimperone have been used successfully in certain animal studies [35], there are no specific 5-HT_2A_ receptor antagonists approved for human use. Nor are there medications capable of increasing the body’s clearance of serotonin.

Most medications currently recommended for the treatment of SS are nonselective receptor antagonists. There is anecdotal evidence to support the use of atypical antipsychotic agents that have 5-HT_2A_ antagonist activity, such as olanzapine and chlorpromazine [10]. However, chlorpromazine also causes sedation while increasing the risk of hypotension and neuroleptic malignant syndrome. The nonselective 5-HT_1A_ and 5-HT_2A_ antagonist cyproheptadine has also been suggested as a potential treatment for SS, with some documented success in the medical literature [95,96]. Based on these findings, cyproheptadine has been added to the list of recommended antidotes to be kept in supply at hospitals in Maryland, per the Maryland Poison Center established within the University of Maryland, School of Pharmacy [97]. Critics of this approach have found no significant differences in outcomes between patients receiving cyproheptadine versus those receiving supportive care alone [12,95,98]

Alternative symptomatic therapies include the beta blocker, propranolol, which possesses 5-HT_1A_ antagonist activity and may ameliorate SS-related tachycardia. However, like chlorpromazine, its use is discouraged due to concerns regarding drug-related hypotension [10,99]. The NMDA receptor antagonist memantine has also shown some promise in rat models of SS, although no current evidence supports its therapeutic efficacy in humans [38].

## 5. Conclusions

The World Health Organization predicts that depression will be the second largest cause of death and disability by 2020, yet up to 70% of patients either do not respond to or do not tolerate their prescribed antidepressant therapy [100]. The solution to this problem likely lies in the pharmacoepigenetic tools of the personalized medicine era [101]. These tools allow us to better understand an individual’s transporter, enzyme, and receptor profile, and to tailor medication therapy (epidrugs) accordingly. Epidrugs could, in theory, alter an individual’s epigenome to lessen the disease burden of depression or to increase an individual’s tolerance to existing antidepressant therapies. This solution is not far off, given that a battery of genes involved in the metabolism of antidepressants as well as the risk of drug-related adverse reactions is already being compiled [100,102]. Furthermore, the FDA has already approved several genetically prescreened drugs in the field of psychiatry [103]. The continuing development of these tools is likely to be our best defense to SS in the setting of a developing epidemic of major depressive disorder in which the most commonly prescribed medications are also the ones that are most likely to cause SS.

## Figures and Tables

**Figure 1 ijms-20-02288-f001:**
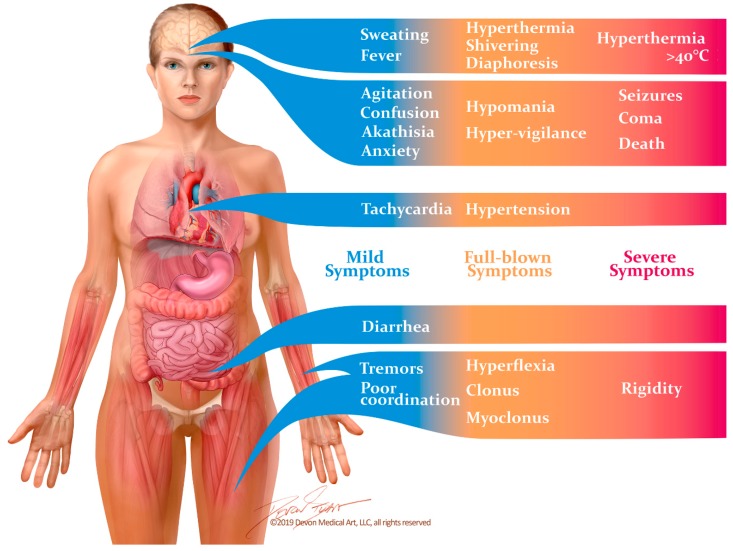
Signs and symptoms of the serotonin syndrome occur along a spectrum of severity. Mild symptoms may easily be overlooked, and may manifest as little more than diarrhea and flu-like symptoms. Unless the disease is recognized and the causative drugs are discontinued, it can rapidly progress to muscle rigidity, severe hyperthermia and death.

**Figure 2 ijms-20-02288-f002:**
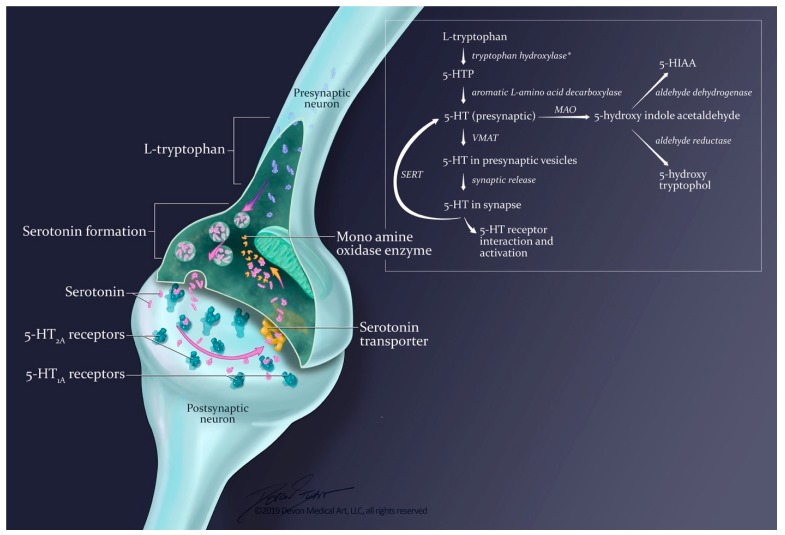
At normal serotonergic neurons, the concentration of serotonin at a synapse is determined by several processes, including synthesis, controlled release from the presynaptic neuron, reuptake, and metabolism. 5-HTP = 5-hydroxytryptophan; 5-HT = 5-hydroxytryptamine (serotonin); VMAT = vesicular monoamine transporter; SERT = serotonin reuptake transporter; 5-HIAA = 5-hydroxy indole acetic acid. Asterisk denotes the rate-limiting step in serotonin synthesis.

**Table 1 ijms-20-02288-t001:** Comparison between the Sternbach, Radomski, and Hunter Criteria for diagnosing serotonin toxicity.

	Sternbach Criteria	Radomski Criteria	Hunter Criteria
**Inclusion Criteria**	Presence of serotonergic medication	Presence of serotonergic medication	Presence of serotonergic medication
**Exclusion Criteria**	Presence of other possible disease etiologies (e.g., infection, substance abuse, and withdrawal) and/or recent addition (or increase in dose) of neuroleptic medication.	None	None
**Signs and Symptoms**	At least three of the following signs/symptoms:	Either four major, or three major plus two minor signs/symptoms:	Any of the following combinations of primary (1°) ± secondary (2°) signs/symptoms:
		Major:	
	Mental status changes (confusion, hypomania)	Impaired consciousness	
		Elevated mood	
	Agitation	Semicoma/coma	1°: Spontaneous clonus alone
		Myoclonus	
	Myoclonus	Tremor	1°: Inducible clonus AND
		Shivering	
	Hyperreflexia	Rigidity	2°: Agitation or diaphoresis
		Hyperreflexia	
	Diaphoresis	Fever	1°: Ocular clonus AND
		Sweating	
	Shivering	Minor:	2°: Agitation or diaphoresis
		Restlessness	
	Tremor	Insomnia	1°: Tremor AND
		Incoordination	
	Diarrhea	Dilated pupils	2°: Hyperreflexia
		Akathisia	
	Incoordination	Tachycardia	1°: Hypertonicity AND fever (temperature >38 °C) AND
		Tachypnea/Dyspnea	
	Fever	Diarrhea	2°: Ocular clonus *or* inducible clonus
		Hypertension/hypotension	

Table adapted from [8,11,14].

**Table 2 ijms-20-02288-t002:** Differential clinical diagnosis for serotonin syndrome.

Disease	Medication Exposure	Shared Clinical Features	Distinguishing Clinical Features
Serotonin Syndrome	Serotonergic medications	Hypertension	Clonus, hyperreflexia Hyperactive bowel sounds
Neuroleptic Malignant Syndrome	Dopamine antagonists	Tachycardia	No clonus or hyperreflexia Bradykinesia
Anticholinergic Toxicity	Acetylcholine antagonist	Hyperthermia	No clonus or hyperreflexia Dry skin Absent bowel sounds
Malignant Hyperthermia	Halogenated anesthetics Succinylcholine	Altered mental Status	No clonus or hyperreflexia Extreme muscular rigidity

Table adapted from [10,20].

**Table 3 ijms-20-02288-t003:** Drugs associated with development of serotonin syndrome, classified according to their mechanism of action.

**Synthesis and Release**			
**Increase Serotonin Synthesis**	Dietary supplements: L-tryptophan		
**Increase Serotonin Release**	Psychostimulants: Amphetamines, phentermine, MDMA Antidepressants: mirtazapine Opioids: meperidine, oxycodone, tramadol Cough suppressants: dextromethorphan		
**Metabolism**			
**Inhibit Serotonin Uptake**	Psychostimulants: Amphetamines, MDMA, cocaine Antidepressants: trazodone SNRI: desvenlafaxine, duloxetine, venlafaxine SSRI: citalopram, escitalopram, fluoxetine, fluvoxamine, paroxetine, sertraline TCA: amitriptyline, amoxapine, clomipramine, desipramine, doxepin, imipramine, maprotiline, nortriptyline, protriptyline, trimipramine Opioids: meperidine, methadone, tramadol Cough suppressants: dextromethorphan		
**Inhibit Serotonin Metabolism**	Anxiolytics: buspirone MAOI: furazolidone, isocarboxazid, linezolid, methylene blue, phenelzine, selegiline, tranylcypromine		
**Inhibit Cytochrome P450 Microsomal Oxidases**	**CYP2D6**	**CYP3A4**	**CYP2C19**
	Inhibitors: fluoxetine, sertraline Substrates: dextromethorphan, oxycodone, risperidone, tramadol	Inhibitors: ciprofloxacin, ritonavir Substrates: methadone, oxycodone, venlafaxine	Inhibitors: fluconazole Substrates: citalopram
**Receptor Activation**			
**Activate Serotonin Receptors**	Hallucinogen: LSD Anxiolytics: buspirone Antidepressants: trazodone Opioids*: fentanyl, meperidine Mood stabilizers: lithium		

* opioids most likely activate serotonergic receptors through a combination of postsynaptic 5-HT receptor stimulation as well as synergistic µ-opioid and 5-HT receptor presynaptic inhibition of GABA release [56]. Adapted from [57].

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
