# Peer review of "The Serotonin Syndrome: From Molecular Mechanisms to Clinical Practice"

_ijms, 2019, doi:10.3390/ijms20092288_

Reviewer 1 Report

In more than one place in  the manuscript the authors state that SS is a "polypharmacy-induced" disorder. This is generally, but not exclusively, the case -- for exampled, tramadol alone has been implicated.

Diagnosis of SS is fairly well covered but a more complete presentation would have included details, including comparisons, of the so-called Radomski criteria. For signs and symptoms, a simple clear table would have sufficed instead of the unnecessary (but visually more striking) artwork. A contrast is the simple, effective and informative presentation of Table 2 dealing with differential diagnosis.

In regard to the pathophysiology of SS, the importance and relevance of  animal models should be considered and, as far as it goes, the coverage is reasonably informative but lacking the extra perspective provided by studies such as those published by Haberzettl et al (2013, 2014) and the Fox group, although some of the latter's work is mentioned under Genetic Polymorphisms.

Under Molecular Pathways, Fig. 2 is once again more artistically impressive than scientifically informative. For those seeking a clearly visualized step-wise sequence of events beginning with the formation of 5-HT from Trp and 5-HTP to vesicular packaging, release, receptor interaction, reuptake via SERT, recycling, MAO-mediated conversion to 5-HTAA, and SERT blockage by drugs, one has to look hard and closely to try and interpret the individual steps. I found the indistinct jumbled artwork of Fig. 2b to be of little use.

In Medications Triggering the SS, tramadol is incorrectly referred to as a phenanthrene opioid. These have a 3-ringed phenanthrene nucleus found in morphine, codeine, buprenorphine, oxycodone etc. In view of accumulating data of the involvement of opioids in SS and the fairly recent FDA drug safety warning concerning the entire class of opioids with SS, opioids are not well covered in the review. This could be obtained by going back to the 2005 Brit J Anaesth review of Gillman and a more up-to-date review by Baldo (2018). The important mechanism study of Tao and Auerbach (2002) is mentioned but only briefly.

Bearing in mind the promise and potential of ongoing genetic polymorphism, SERT and receptor studies, the Conclusion is of little interest, being essentially a few brief re-hashed statements gleaned from the Abstract, Introduction and a few sections of text. Its content should be upgraded and more forward looking (preferably) or deleted.

The authors are inconsistent in their presentation of article titles in that upper case is sometimes used instead of lower case -- see for example references 9, 10, 34, 63 64, 69 etc.  In addition, references 73 and 79 are the same and reference 74 is unnecessary.

Author Response

Response to Comments from Reviewer #1

In more than one place in  the manuscript the authors state that SS is a "polypharmacy-induced" disorder. This is generally, but not exclusively, the case -- for example, tramadol alone has been implicated.

We have incorporated the reviewer’s suggestion and edited our manuscript accordingly. We have emphasized that while polypharmacy is the most common cause of SS (page 8), it is not the exclusive cause.

Diagnosis of SS is fairly well covered but a more complete presentation would have included details, including comparisons, of the so-called Radomski criteria. For signs and symptoms, a simple clear table would have sufficed instead of the unnecessary (but visually more striking) artwork. A contrast is the simple, effective and informative presentation of Table 2 dealing with differential diagnosis.

We thank the reviewer for the suggestion, and we have edited Table 1 accordingly. We believe that the value of Figure 1 lies more in a visual representation that SS may manifest as a wide spectrum of symptoms, rather than to serve as a comprehensive description of the signs and symptoms of this disorder. The figure legend (and one of the key messages we wish to convey in this Review) emphasizes the need for vigilance to stop this preventable complication from resulting in death.

In regard to the pathophysiology of SS, the importance and relevance of animal models should be considered and, as far as it goes, the coverage is reasonably informative but lacking the extra perspective provided by studies such as those published by Haberzettl et al (2013, 2014) and the Fox group, although some of the latter's work is mentioned under Genetic Polymorphisms.

We thank the reviewer for the information provided to help us improve the quality of this review, and we have expanded the section on animal models per the reviewer’s suggestion.

Under Molecular Pathways, Fig. 2 is once again more artistically impressive than scientifically informative. For those seeking a clearly visualized step-wise sequence of events beginning with the formation of 5-HT from Trp and 5-HTP to vesicular packaging, release, receptor interaction, reuptake via SERT, recycling, MAO-mediated conversion to 5-HTAA, and SERT blockage by drugs, one has to look hard and closely to try and interpret the individual steps. I found the indistinct jumbled artwork of Fig. 2b to be of little use.

We have edited the figure by removing figure 2b and elaborating figure 2a (now “Figure 2”) to include a stepwise diagram of serotonin metabolism.

In Medications Triggering the SS, tramadol is incorrectly referred to as a phenanthrene opioid. These have a 3-ringed phenanthrene nucleus found in morphine, codeine, buprenorphine, oxycodone etc. In view of accumulating data of the involvement of opioids in SS and the fairly recent FDA drug safety warning concerning the entire class of opioids with SS, opioids are not well covered in the review. This could be obtained by going back to the 2005 Brit J Anaesth review of Gillman and a more up-to-date review by Baldo (2018). The important mechanism study of Tao and Auerbach (2002) is mentioned but only briefly.

We have taken the reviewer’s suggestion and synthesized these studies into two additional paragraphs on page 11.

Bearing in mind the promise and potential of ongoing genetic polymorphism, SERT and receptor studies, the Conclusion is of little interest, being essentially a few brief re-hashed statements gleaned from the Abstract, Introduction and a few sections of text. Its content should be upgraded and more forward looking (preferably) or deleted.

We thank the reviewer for the suggestion and we have re-written the conclusion to address the role of pharmacoepigenetics in management of this disease.

The authors are inconsistent in their presentation of article titles in that upper case is sometimes used instead of lower case -- see for example references 9, 10, 34, 63 64, 69 etc. In addition, references 73 and 79 are the same and reference 74 is unnecessary.

We thank the reviewer for pointing out this oversight. The Gillman references have been consolidated, and the upper/lower case inconsistencies have been corrected. Please note that the references now have different numbers due to manuscript  editing (and the addition of references) in response to feedback provided by the journal reviewers.

Reviewer 2 Report

In this paper, the authors reviewed the diagnosis, pathogenesis and treatment of serotonin syndrome. I have the following comments for the authors.

 1.  For “Differential clinical diagnosis for the Serotonin Syndrome”, it needs to include antidepressant discontinuation syndrome and alcohol/BZD withdrawal syndrome.

2.  For “Neuroleptic malignant syndrome is typically associated with the use of neuroleptic medications”, it’s better to use “antipsychotic medications” than “neuroleptic medications”.

3. Table 3. Please make sure that phentermine and cocaine can increase serotonin synthesis.

4. For “at the T102C site of the 5-HT2A receptor”, Please add “gene” after “5-HT2A receptor”.

5.  Are there any reported risk factors associated with the development of serotonin syndrome? Such as age, gender.

6. The authors failed to cite a related review report (Demystifying serotonin syndrome (or serotonin toxicity) Can Fam Physician 2018;64:720).

Author Response

Response to Comments from Reviewer #2:

In this paper, the authors reviewed the diagnosis, pathogenesis and treatment of serotonin syndrome. I have the following comments for the authors.

 1.  For “Differential clinical diagnosis for the Serotonin Syndrome”, it needs to include antidepressant discontinuation syndrome and alcohol/BZD withdrawal syndrome.

We thank the reviewer for the suggestion and we have edited the manuscript accordingly by elaborating on antidepressant discontinuation syndrome and alcohol/BZD withdrawal syndrome in the body of the text. We still felt that NMS, anticholinergic toxicity and MH were most closely related to SS, and so for the sake of simplicity we only included these threesyndromes  in Table 2.

2.  For “Neuroleptic malignant syndrome is typically associated with the use of neuroleptic medications”, it’s better to use “antipsychotic medications” than “neuroleptic medications”.

We thank the reviewer for the suggestion and we have edited the manuscript accordingly.

3. Table 3. Please make sure that phentermine and cocaine can increase serotonin synthesis.

The reviewer is correct to make this point. We checked the reference used by the original source and found the evidence to be weak. Therefore, we edited the table so as to remove cocaine, phentermine, and amphetamines from causes of increased serotonin synthesis

4. For “at the T102C site of the 5-HT2A receptor”, Please add “gene” after “5-HT2A receptor”.

We thank the reviewer for the suggestion and we have edited the manuscript accordingly.

5.  Are there any reported risk factors associated with the development of serotonin syndrome? Such as age, gender.

Despite an extensive literature search, we were not able to find any reported risk factors associated with the development of serotonin syndrome except for the use of medications that can trigger the syndrome. While age and gender have not been found to be related to the incidence of or predisposition to serotonin syndrome, many of the medications with the potential to cause this condition are commonly used in the geriatric population. This has been added to the revised manuscript.

6. The authors failed to cite a related review report (Demystifying serotonin syndrome (or serotonin toxicity) Can Fam Physician 2018;64:720).

We thank the reviewer for the suggestion and we have edited the manuscript accordingly in order to include this reference.

Reviewer 3 Report

The authors mentioned the mechanisms of Serotonin syndrome. This is the important aspect of the treatment. Because the hypothesis and fact are complicated, the author should clarify and add more references.

1.      Fig 1 and 2) Please change color. This is not kind for color blindness like me.

2.      Table 2) What is the definition of neuromuscular excitation? Extreme muscular rigidity may be included in neuromuscular excitation. Please change anticholinergic delirium to anticholinergic toxicity.

3.      Table 3) Because the authors distinguish the fact and hypothesis, you should show each reference that each medication, such as amphetamine, cocaine, L-tryptophan, MAOI, SSRI …etc. induces Serotonin syndrome. Amphetamine, cocaine, MDMA, phentermine, should be defined as psychostimulant.

4.      CYPs) Are there any reference which indicate the combination therapy of psychotropic medication induce Serotonin syndrome more?

Author Response

Response to Comments from Reviewer #3:

1.      The authors mentioned the mechanisms of Serotonin syndrome. This is the important aspect of the treatment. Because the hypothesis and fact are complicated, the author should clarify and add more references.

We thank the reviewer for the comments, and we have elaborated on the mechanisms of serotonin syndrome with opioids (page 11). However, as noted in the manuscript, there is a dearth of mechanistic evidence regarding SS. Several potential mechanisms (and the relevant evidence) have been presented in 9-12.

2.      Fig 1 and 2) Please change color. This is not kind for color blindness like me.

 We appreciate this feedback and we re-made the figures to increase the contrast between the colors -  this will hopefully make the figures easier to see.

 3.      Table 2) What is the definition of neuromuscular excitation? Extreme muscular rigidity may be included in neuromuscular excitation. Please change anticholinergic delirium to anticholinergic toxicity.

 We thank the reviewer for the suggestion and we have edited Table 2 accordingly.

 4.      Table 3) Because the authors distinguish the fact and hypothesis, you should show each reference that each medication, such as amphetamine, cocaine, L-tryptophan, MAOI, SSRI …etc. induces Serotonin syndrome. Amphetamine, cocaine, MDMA, phentermine, should be defined as psychostimulant.

 We thank the reviewer for the suggestion and we have edited Table 2 according to the suggestions. Most drugs listed in the table have been referenced more comprehensively in the text. Due to the amount of information already contained in the table, we believe that addition of further information and references within the table itself may detract from the value of the table.

 5.      CYPs) Are there any reference which indicate the combination therapy of psychotropic medication induce Serotonin syndrome more?

 We have researched this and found that the combination of an MAOI with an SSRI, and SNRI or another MAOI is the most dangerous combination and most likely to result in SS (Foong et al., 2018). We have included this information into the text (page 8). Food and Drug Administration data safety reporting confirms that MAOI in combination with other serotonergic medications is most likely to result in SS.

  Round  2

Reviewer 3 Report

none

Author Response

First, the category of the drugs was (and still is) sometimes questionable.

We thank the reviewer for this comment. We intend the table to be primarily a cognitive tool used to provide a visual depiction of different mechanisms by which SS may occur. This could guide clinicians to identify drugs which may trigger serotonergic effects, thereby benefiting patients. We intentionally tried to keep the table as simple as possible, rather than being a comprehensive list of references and mechanisms related to each drug, in order to achieve our intended goal.

- LSD cannot be considered as a psychotimulant; it would better correspond to a “hallucinogenic drug” or a “dissociative agent”

We have made the requested change.

- As far as I know, mirtazapine is an antagonist or inverse agonist at several 5-HT receptors, not an agonist.

We refer the reviewer to page 12 of the manuscript, which contains clarification regarding this point. We have edited the table accordingly. While mirtazapine is not a serotonin agonist, it nonetheless increases serotonergic transmission.

- Amphetamines are poorly effective in enhancing 5-HT release simply because their affinity for the SERT is very low.

We thank the reviewer for this comment, and while we agree that amphetamines effect serotonergic neurons in different ways, a review of the recent literature confirms the concepts of amphetamine-induced increases in serotonin release and inhibition of serotonin reuptake, as identified in table 3. We feel that an extensive discussion regarding the multiple effects of amphetamines on serotonin neurons was beyond the scope of this Review article.

- L-DOPA would rather decrease 5-HT extracellular levels (De Deurwaerdère et al. 2017 for review).

We thank the reviewer for identifying this error. We have updated the table accordingly

- Receptor action by opioids should be indirect. Note that the ability of morphine to enhance 5-HT extracellular levels occurred at quite elevated dose (5 to 10 mg/kg in rats; the studies by Tao and Auerbach)

We thank the reviewer for this information, and we agree that indirect receptor stimulation is a major mechanism of action of serotonergic stimulation (we have added a footnote to the table for clarification). However, we contend that this is not the only mechanism reported in the literature. We refer the reviewer to the paper by Baldo et al., 2018.

- The inhibition of SERT by ondansetron or granisetron is not evident at clinical doses because their affinity is in the micromolar range

We have removed these from the table as they are not referenced in the text.

The message here is that you should report in this table only the data for which you are sure, eventually associated with appropriate references. You could also discard some entries in this table that are not evoked in the text.

We have followed the reviewer’s suggestion and consolidated the table entries to match those only mentioned in the text.